

# Outline of an experimental design aimed to detect protein A mirror image in solution

Osvaldo A. Martin[1], Yury Vorobjev[2], Harold A. Scheraga[3] and Jorge A. Vila[1,3]

[1] Instituto de Matemática Aplicada San Luis, UNSL-CONICET, San Luis, Argentina
[2] Institute of Chemical Biology and Fundamental Medicine, Siberian Branch of the Russian Academy of Science, Novosibirsk, Russia
[3] Baker Laboratory of Chemistry and Chemical Biology, Cornell University, Ithaca, NY, United States of America

## ABSTRACT

There is abundant theoretical evidence indicating that a mirror image of Protein A may occur during the protein folding process. However, as to whether such mirror image exists in solution is an unsolved issue. Here we provide outline of an experimental design aimed to detect the mirror image of Protein A in solution. The proposal is based on computational simulations indicating that the use of a mutant of protein A, namely Q10H, could be used to detect the mirror image conformation in solution. Our results indicate that the native conformation of the protein A should have a pKa, for the Q10H mutant, at $\approx$6.2, while the mirror-image conformation should have a pKa close to $\approx$7.3. Naturally, if all the population is in the native state for the Q10H mutant, the pKa should be $\approx$6.2, while, if all are in the mirror-image state, it would be $\approx$7.3, and, if it is a mixture, the pKa should be larger than 6.2, presumably in proportion to the mirror population. In addition, evidence is provided indicating the tautomeric distribution of H10 must also change between the native and mirror conformations. Although this may not be completely relevant for the purpose of determining whether the protein A mirror image exists in solution, it could provide valuable information to validate the pKa findings. We hope this proposal will foster experimental work on this problem either by direct application of our proposed experimental design or serving as inspiration and motivation for other experiments.

# INTRODUCTION

A mirror image conformation is one that looks approximately like the specular image of the native state. We say approximately because we do not require the amino acids to be specular images, but only the overall topology of the molecule. At least for some proteins, the mirror image will be energetically very close to the native state and thus it could also exist in solution. Among these proteins, we will focus our attention on the B-domain of staphylococcal protein A [PDB ID: 1BDD; a three–helix bundle] (*Gouda et al., 1992*). This protein has been the subject of extensive theoretical (*Olszewski, Kolinski & Skolnick, 1996*; *Vila, Ripoll & Scheraga, 2003*; *Garcia & Onuchic, 2003*; *Lee et al., 2006*;

Corresponding authors
Osvaldo A. Martin,
omarti@unsl.edu.ar
Jorge A. Vila, jv84@cornell.edu

*Kachlishvili et al., 2014*) and experimental (*Deisenhofer, 1981*; *Gouda et al., 1992*; *Bai et al., 1997*; *Myers & Oas, 2001*; *Sato et al., 2004*; *Dimitriadis et al., 2004*; *Noel et al., 2012*) studies because of its biological importance and small size. In contrast to this, the mirror-image conformation has been subject of limited discussion (*Olszewski, Kolinski & Skolnick, 1996*; *Vila, Ripoll & Scheraga, 2003*; *Garcia & Onuchic, 2003*; *Noel et al., 2012*; *Kachlishvili et al., 2014*). The reason for this might be that the mirror image conformation of this protein has been observed only in some theoretical studies with different force fields but it has never been detected experimentally. As to whether this conformation is an artifact of the simulations or is difficult to observe the conformation experimentally, remains to be solved.

Difficulties for experiments to detect the mirror-image conformation arise precisely because the secondary structures of the mirror-image and the native conformation of protein A are identical and the structural difference between these conformations are subtle (*Kachlishvili et al., 2014*). Because of this, use of simple experiments such as circular dichroism, used to estimate the fraction of secondary-structure content, or more sophisticative technique, such as nuclear magnetic resonance (NMR) spectroscopy, e.g., to monitor the $^{13}C$ chemical shift changes that may occur at residue-level (*Kachlishvili et al., 2014*), are useless for an accurate characterization of the mirror image conformation. A strong motivation to propose alternative methods to explore the possible coexistence in solution of the native and mirror-image conformation of protein A, comes from older evidence indicating that the mirror-image conformation could be a possible solution to the NMR-determined structure of protein A (*Gouda et al., 1992*). Indeed, according to *Gouda et al. (1992)*, "…*distance-geometry calculations resulted in 41 solutions, which had correct polypeptide folds excluding 14 mirror-image substructures …*" However, the mirror-image structures were excluded from the analysis of *Gouda et al. (1992)* without providing any reason. It seems that the decision was adopted because the mirror-image satisfies the NOE constraints but contain D-amino acid residues (Jorge Vila, Ichio Shimada, pers. comm., 2015).

Overall, we propose here a proof-of-concept of an experimental design aimed to solve this problem. Initially we will show, by using ROSETTA, (*Bradley, Misura & Baker, 2005*) that a mutant of protein A, hereafter the Q10H protein, exhibits the ability to fold into the native conformation (see Fig. 1) as well as into the mirror-image conformation (see Fig. 2). Later, we estimate the fraction of the native and mirror-image populations of the protein Q10H by using a recently introduced method, that take into account the protein dynamics in water by using a constant-pH MD simulation, to accurately determine the pKa values of ionizable residues, and fractions of ionized and tautomeric forms of histidine (His), in proteins at a given fixed pH (*Vorobjev, Scheraga & Vila, 2018*). Indeed, we explore the dependence of the electrostatically-calculated pKa and fractions of the imidazole ring forms of H10 as a function of pH for both the native-like and mirror-image conformations.

## MATERIALS AND METHODS

In this section we will give a brief reference to existent theoretical methods aimed to predict (i) the 3D structure of proteins accurately; *Bradley et al. (2003)* or determine (ii) the pKa

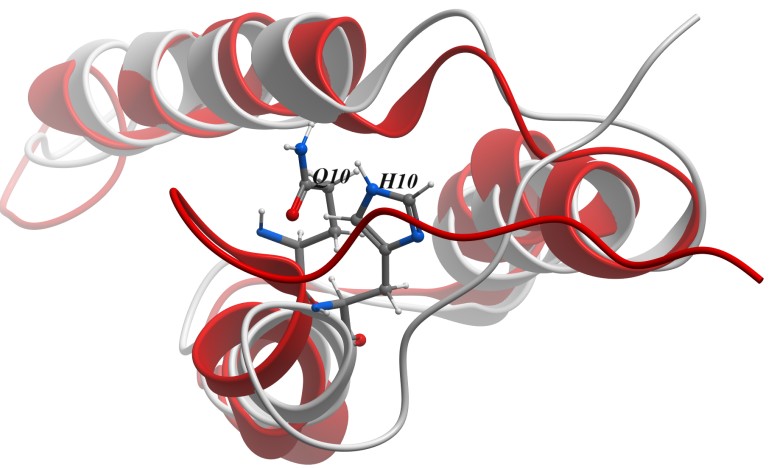

**Figure 1** **Red- and white-ribbon diagrams for the native structures of protein A (PDB ID 1BDD** *Gouda et al., 1992***) and the equivalent for protein Q10H, respectively.** The position of the side-chain of Q10 and H10 for protein A and protein Q10H are highlighted. The $C^{\alpha}$ rmsd between the two native structures is 1.4 Å.

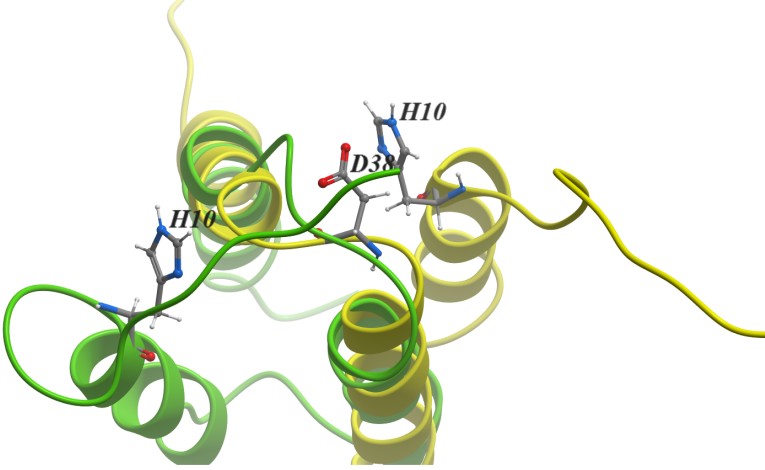

**Figure 2** **Green- and yellow-ribbon diagrams for the native and "mirror" image conformations of Protein A, respectively.** The position of the side-chain of H10 is highlighted for each of these conformations. Moreover, the side-chain of D38 is also displayed to point out the close proximity between D38 and H10 in the mirror image conformation. The favorable electrostatic interaction between D38 and H10 may be responsible for the large ($\Delta \approx -1.1$) change in the computed pKa between the native-like and the mirror-image conformations.

values of ionizable residues and fractions of ionized and tautomeric forms of histidine (His) and acid residues in proteins, at a given fixed pH (*Vorobjev, Vila & Scheraga, 2008*; *Vorobjev, Scheraga & Vila, 2018*).

## Determination of the native and image-mirror conformations of protein Q10H

To generate the native and mirror-image conformations of protein A Q10H, we used the fast-relax protocol from Rosetta (*Chaudhury, Lyskov & Gray, 2010*; *Bradley, Misura & Baker, 2005*); this is an all-atom refinement protocol consisting of several rounds of repacking and energy minimization. The repulsive part of the Van der Waals energy function is annealed from 2% to 100%. Essentially the algorithm explores the local conformational space around the starting structure with a radius of 2 to 3 Å of rmsd (for the $C^{\alpha}$). We performed several rounds of fast-relax using the following genetic-like algorithm:

1. For a given conformation of protein A mutate it by replacing Q10 with H10;
2. Use the mutant as the starting point of 200 independent rounds of the fast-relaxation protocol;
3. Choose 10 conformations; 2 at random and the 8 lowest-energy conformations;
4. For each one of those conformations use fast-relaxation to generate 100 independent rounds (for a total of 1,000 conformations);
5. repeat, from step 3, 40 times;
6. keep the lowest energy conformation from all the rounds.

We started from two different conformations. For the native conformation we used 1BDD (*Gouda et al., 1992*). For the mirror image, we started from a mirror-image conformation previously obtained by *Vila, Ripoll & Scheraga (2003)*.

The Rosetta energy score of the lowest energy conformations for the native and image-mirror of protein Q10H was on par.

## Computation of the pKa and the tautomeric fractions of the imidazole ring of H10

The native-like and mirror-image conformations of protein Q10H, generated as describe in the previous section, were used as input files for the calculations of the pKa of all ionizable residues in the sequence as well as the fractions of the ionized $H^{+}$ and the tautomeric $N^{\epsilon 2} - H$ and $N^{\delta 1} - H$ forms of the imidazole ring of H10. In particular, as it is well known, the tautomeric determination of the imidazole ring of His is both a very important problem in structural biology (*Schnell & Chou, 2008*; *Bermúdez et al., 2014*) and a challenging task (*Machuqueiro & Baptista, 2011*). For this reason, a recently introduced electrostatic-based method to determine the pKa values of ionizable residues and fractions of ionized and tautomeric forms of histidine (His) and acid residues in proteins (*Vorobjev, Scheraga & Vila, 2018*), is applied here to the analysis of protein A mutant Q10H. Protein dynamics in water, at a given pH = 7.0, was taken into account by constant-pH MD simulation (*Vorobjev, Scheraga & Vila, 2017*; *Vorobjev, Scheraga & Vila, 2018*) of both the native and mirror-image conformations of the Q10H mutant.

Protein dynamics in water was modeled by MD simulations with implicit solvent, namely using the Lazaridis–Karplus solvent model (*Lazaridis & Karplus, 1999*) with the BioPASED program (*Popov & Vorobev, 2010*). For the MD simulation, the following three-step protocol was used. First step, determination of an equilibrium protein structure
at temperature 300 K and pH 7.0 using the next three step procedure: (i) building a full atomic protein structure, i.e., with all hydrogen atoms added; this means, for example, that each His residue needs to be built up in the most probable form, i.e., in the ionized $H^+$ form or in the most probable neutral tautomer, $N^{\delta 1} - H$ (HD1 or HID) and $N^{\epsilon 2} - H$ (HE2 or HIE); (ii) the crystal structure with all the assigned hydrogen atoms and histidine forms was energy optimized in implicit solvent using a conjugate gradient method; (iii) the system is heated slowly from 1 to 300 K during 250 ps; and (iv) a final equilibration at 300 K, during 0.5–1 ns, was carried out. Step 2: generation of a representative set of 3D protein structures as a collections of snapshots each 50 ps along equilibrium MD trajectory during 25 ns snapshots taken every 50 ps time-interval. Step 3: for each snapshot, the pKa's of all ionizable residues is computed, as well as the fractions of two neutral tautomers of His and the acid residues, by carrying out an MC calculation with GB-MSR6c as an implicit solvent model. Finally, an average pKa's for each ionizable residue as well as the fraction of ionized and two tautomers of histidine and neutral form of acid residues of the protein are calculated.

The ionization constants pKa and the fractions of ionized and two neutral tautomers of histidine at constant pH 7.0 are modeled by MD simulations at constant pH (*Vorobjev, Scheraga & Vila, 2017*; *Vorobjev, Scheraga & Vila, 2018*). During the pH-constant MD simulations all acid (Asp, Glu) and base (Lys, Arg) residues were kept in the ionized state because their respective pKo's (3.5, 4.0, and 10.5, 12.5, respectively) are shifted by more than 2.5 pK units from the pH (7.0) at which the calculations were carried out (see Table S1). On the other hand, the two existent histidine residues, namely H10 and H19, were considered to be electrostatically couple residues having nine ionization states, namely, 00, 01, 02, 10, 11, 12, 20, 21, 22, where 0,1,2 represents the ionized and two neutral tautomer states respectively (see Table S2). The average potential energy values and it's thermal fluctuations due to molecular dynamics in solvent are estimated along 25 ns MD equilibrium trajectory for each of the nine ionization states (see Table S2). Low energy states, which have occupation number large than 0.01, for histidine residues H10 and H19 along 25 ns constant-pH MD trajectory, are shown in Table S3. Energy fluctuations of the Q10H protein in solvent along 25 ns MD trajectory for native-like and mirror-image structures are shown in Fig. S1. It can be seen, from this figure, that fluctuation of the native-like and mirror image structures are overlapping, i.e., spontaneous transition between native-like and mirror-image structures can occur. The average range of fluctuations of the atomic positions, i.e., in terms of the RMSD, observed along the MD trajectories were 1.4 and 1.3 Å for the native-like and mirror-image structures, respectively. Variation of pKa constant along MD trajectory is presented on Fig. S2. It can be seen that pKa shift for histidine His10 are −0.3 and +0.8 pK units for the native-like and mirror-image protein structures. Such relatively large pKa shift for relatively small proteins can serves as a mark of native-like and mirror-image structures. Occupations of ionization states of His10 residue versus MD time are shown in Figs. S3A and S3B for native-like and mirror-mage structures, respectively. It should be noticed that occupation of different ionization states of His10 show a large variation, i.e., RMSD from it's average values.

One challenge question is how meaningful the pK difference computed with our method are. In this regard, we would like to mention that the accuracy of the pK calculations have been carefully analyzed through a series of applications. Indeed, a comparison with experimental data show the method is accurate enough, in terms of a NMR-based methodology, to predict the pK and tautomeric fractions of six histidine forms on the enzyme DFPase from *Loligo vulgaris*, a 314-residues all-$\beta$ protein containing 94 ionizable residues (*Vorobjev, Scheraga & Vila, 2018*). In addition, a large test on 297 ionized residues from 34 proteins show that a 57%, 86% and 95% of the pK prediction are with an accuracy better than 0.5, 1.0 and 1.5 pK unit respectively (*Vorobjev, Scheraga & Vila, 2017*). Such range of accuracy is comparable or better than state of the art predictive methods such as the electrostatic-based MCCI2 method (*Song, Mao & Gunner, 2009*).

Moreover, the H10 pKa differences between the native-like and the mirror-image conformations of Q10H protein does not disappear but kept constant ($\approx$1.1 pK units) between 9 ns–25 ns of the pH-constant MD simulation (see Fig. S2 and Table S3), hence given further confidence on the accuracy of the pK shift predictions.

Since the error of our pKa predictive method can be positive or negative, there is a non-negligible chance that the computed difference of ($\approx$1.1 pK units) is in fact practically null. However, there is also a non-negligible chance that the pKa difference is even larger than 1.1 pK units. Thus, we hope this result are enough to encourage experimentalists to perform the experimental design we propose.

## RESULTS AND DISCUSSION

### Analysis of the pKa variations as a function of pH

Figure 2 shows a superposition of the lowest-energy conformations for both the native-like (green-ribbon) and the mirror-image (yellow-ribbon) of protein Q10H obtained by using ROSETTA (*Bradley, Misura & Baker, 2005*). These two structures were used to compute for each ionizable residue along the sequence the value of the pKa variations ($\Delta = [pK^a_{Native} - pK^a_{Mirror}]$) at pH 7.0 (*Vorobjev, Vila & Scheraga, 2008*). The result of this analysis is shown in Fig. 3 (as blue dots) where one of the largest change in $\Delta$, namely larger than $\pm$1.0 pK units, occurs for H10. This large shift on the pKa of residue H10 appears to be a consequence of the close proximity of H10 to D38 in the mirror-image conformation of protein Q10H (see Fig. 2).

There is another change of $\Delta$ larger than $\pm$1.0 pKa unit and it occurs for residue K8 (see blue dots in Fig. 3), a residue belonging to the flexible N-terminal region of the mutant protein Q10H, viz., ranging from residues T1 trough E9. The origin of the large computed shift in the pKa of residue K8 is the following. In the native structure of protein Q10H residue K8 is well exposed to the solvent. On the other hand, in the mirror image of Q10H residue K8 is close to E16, making a favorable electrostatic interaction. However, a close inspection of these two structures indicates that the favorable electrostatic-interaction between K8 and E16, observed in the mirror image conformation, could also occur on the native conformation, e.g., by a rearrangement of the backbone-torsional angles of the flexible N-terminal region of the protein Q10H. If this were feasible, the computed pKa

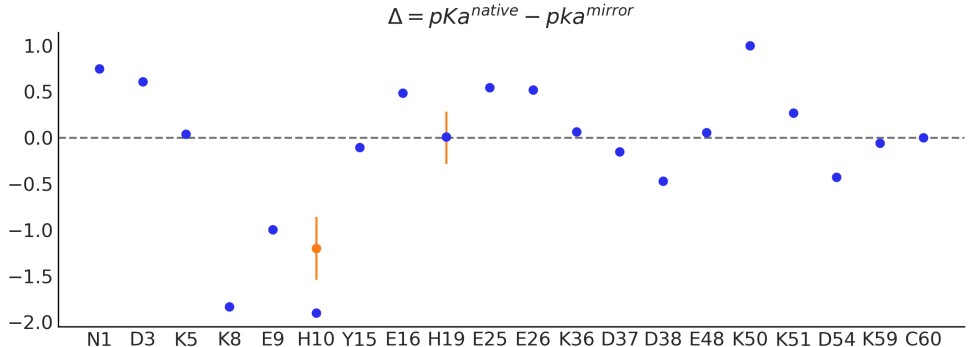

**Figure 3** **Dots indicate the pKa change (Δ), computed at pH 7.0, for each ionizable residue along the protein Q10H sequence.** The blue-dots were computed from the single lowest-energy generated conformations of both the native-like and mirror-image topology, respectively. The orange-dots were computed for the two histidines in the sequence, namely H10 and H19, as an average over 25 ns MD simulations for both the native-like and mirror-image conformations; vertical orange-lines denotes the standard deviations of the computed average Δ values.

shift for K8 should be $\approx 0$. Consequently, monitoring the pKa shift of K8 does not appear to be the right choice for the purpose of an accurate determination of the coexistence between the native and the mirror image states in solution. Unlike the origin of the pKa shift for K8, the interaction between H10 and D38 cannot take place in both the native and the mirror-image conformations (see Fig. 2) and, hence, from here on we will focus our attention on H10 only.

Consideration of the protein dynamics in water is very important for an accurate computation of conformational-dependent values, such as the pKa's. However, this effect was not taken into account in the computation of the Δ values shown, as blue-dots, in Fig. 3. Consequently, we carried out a constant-pH MD simulation of both the native-like Q10H mutant and its mirror-image conformations (*Vorobjev, Scheraga & Vila, 2018*). As mentioned in the 'Materials and Methods', during the simulations at constant-pH 7.0 it is reasonable to consider all acid (Asp, Glu) and base (Lys, Arg) residues in the ionized state, because their respective $pK_0$'s (3.5, 4.0, and 10.5, 12.5, respectively) are shifted by more than 2.5 pK units from the pH (7.0) at which the calculations were carried out. For the same reason, the only Tyr in the sequence was consider as neutral. However, histidine residue pKa's (6.5) can vary considerably at pH 7.0 at which the calculations are carried out and, hence, consideration of histidine ionization states for each of the imidazole ring of His forms must be considered explicitly. Consequently, during the calculations the nine ionizations states of the two interacting His, namely between H10 and H19, were explicitly considered (see Table S2). The average Δ change for H10, computed from the native-like and mirror-image conformations after 25 ns MD simulations, is shown as an orange dot in Fig. 3. Similarly, the computed average change for the imidazole ring forms of H10 as a function of pH for both the native like and the mirror image conformations are displayed in Fig. 4. As shown in Fig. 4 at a given fix pH, e.g., at pH = 8.0, there are significant changes among the computed fractions of the imidazole ring forms of H10.
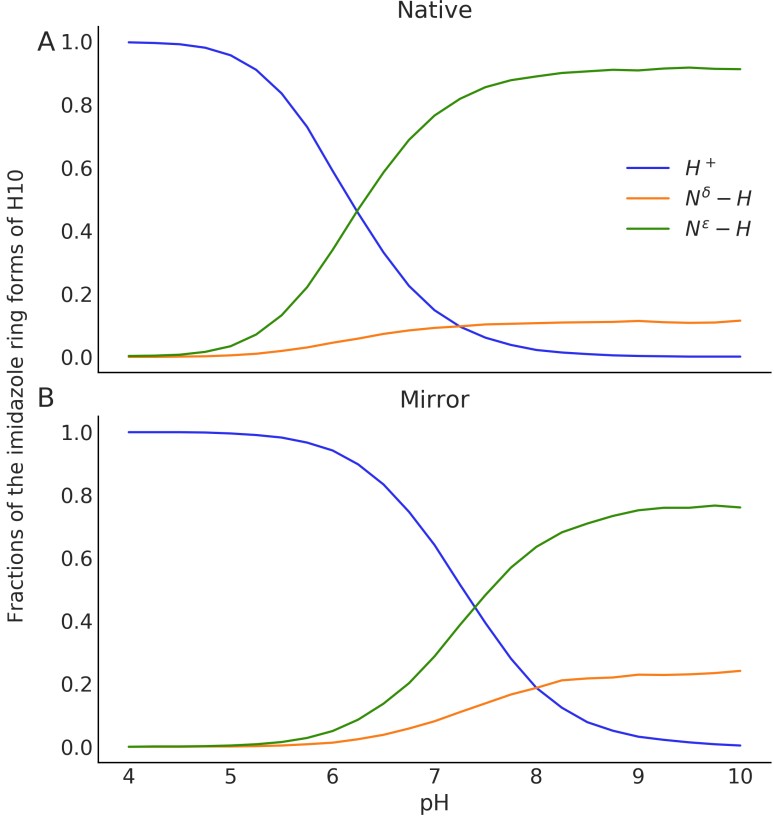

**Figure 4** **Fractions of the imidazol ring forms of H10 as a function of pH, for the "Native" (A) and "Mirror" (B) topologies of the Q10H mutant of protein A.** The values, for each topology, are estimated along 25 ns MD equilibrium trajectory for each of nine ionization states of two electrostatically-coupled histidines residues, namely H10 and H19.

In general, the results shown in Figs. 3 and 4 and Table S3) are decisive for the determination of the fraction of native and mirror image conformations in solution. Indeed, if the dominant conformation in solution is the native like then the pKa of H10 will be $6.2 \pm 0.2$. On the other hand, if the dominant conformation in solution is the mirror image then the pKa will be $7.3 \pm 0.2$. Any other in-between value may indicate coexistence of these two conformations in solution.

## Validation of the H10 pKa-based predictions

Small changes around the computed average pKa value for H10 in the native-like conformation (6.2) are of course possible. In such a case additional experiments are necessary to determine whether such shift is due to expected fluctuations of the native conformation (around $\pm 0.3$ in pKa units) or to the presence of a small fraction of the mirror-image conformation. One such additional experiment could be the determination of the tautomers of the imidazole ring of H10. In this section we analyze this possibility by using two NMR-based methods.

First, as shown in Fig. 4, there is a large change in the average fractions of H10 tautomers as a function of pH. In particular, if the population of the native-like conformation is dominant in solution ($\approx$100%) then, as shown in Fig. 4, the fraction of the protonated form should be $\approx 0$% at pH $\approx 8.0$. In other words, only the imidazole ring of H10 tautomers will be present in solution at this pH. Therefore, their relative populations can be determined accurately by measuring the one-bond CH, $^1J_{CH}$, Spin-Spin Coupling Constants (SSCC) of the imidazole ring of H10.

Let us explain this in detail. Under the only condition that His is non-protonated, we have been able to show that the fraction of the $N^{\delta 1} - H$ tautomeric form ($f^{\delta 1}$) of the imidazole ring of His can be estimated by using the following equation: $f^{\delta 1} = (J^{obs} - 165.0)/15.0$, (Vila & Scheraga, 2017) where $J$ refers to $^1J_{C\delta 2H}$ SSCC, and here obs is the observed value in solution for H10. Naturally, $f^{\epsilon 2} = 1 - f^{\delta 1}$. Hence, if the native-like structure is the dominant topology in solution, then the following inequality should hold: $f^{\epsilon 2} \gg f^{\delta 1}$ (see Fig. 4) otherwise there would be coexistence of the native-like structure with other topology in solution.

A second, and less restrictive, validation test will be to use a recently proposed NMR-based methodology aimed to determine the tautomeric forms as a function of the ionization state of the imidazole ring of histidine (Vila et al., 2011). In this approach, the average tautomeric fraction of the $N^{\epsilon 2} - H$ form of His ($f^{\epsilon 2}$) can be determined by using the following equation: $f^{\epsilon 2} = \Delta^{obs}(1 - f^{H+})/\Delta^{\epsilon}$ where $f^{H+}$ is the experimentally determined fraction for the ionized form of H10, at a given fix pH; $\Delta^{obs} = |^{13}C^{\delta 2} - ^{13}C^{\gamma}|$, where $^{13}C^{\delta 2}$ and $^{13}C^{\gamma}$ are the NMR-observed chemical shifts for the imidazole ring of H10 at that pH; and $\Delta^{\epsilon}$ is the first-order absolute shielding difference, $|^{13}C^{\delta 2} - ^{13}C^{\gamma}|^{\epsilon}$, between the $^{13}C^{\delta 2}$ and $^{13}C^{\gamma}$ nuclei for the $N^{\epsilon 2} - H$ tautomer, i.e., present to the extent of 100%. $\Delta^{\epsilon}$ is a parameter which must be estimated (Vila et al., 2011). As a first approximation, a $\Delta^{\epsilon} = 27.0ppm$, obtained from the analysis of a His-rich protein, (Vorobjev, Scheraga & Vila, 2017) namely Loligo vulgaris (pdb id 1E1A), a 314-residue all-$\beta$ protein, (Scharff et al., 2001) should be used. Naturally, the $f^{\delta 1}$ fraction, viz., for the $N^{\delta 1} - H$ tautomer, is obtained straightforwardly as: $f^{\delta 1} = 1 - f^{H+} - f^{\epsilon 2}$. Although this second approach to compute the tautomers of H10 it is more general than the previous one, i.e., by using the $^1J_{C\delta 2H}$ SSCC, the determination of the $^{13}C^{\gamma}$ chemical shift it is not always feasible. Indeed, only 213 $^{13}C^{\gamma}$, versus 6,984 $^{13}C^{\delta 2}$, chemical shifts of the imidazole ring of histidine have been deposited in the Biological Magnetic Resonance data Bank (BMRB) (Ulrich et al., 2008). Overall, if it were feasible to observe the $^{13}C^{\gamma}$ chemical shift, we suggest that both approaches should be used to validate the pKa predictions.

Although this work is not intended to be a revision of all existing methods used to determine the tautomeric forms of the imidazole ring of His, the use of the tautomeric identification by direct observation of $^{15}N$ chemical shifts of the imidazole ring of His, which is a common practice in NMR spectroscopy, (Pelton et al., 1993; Shimahara et al., 2007; Hass et al., 2008) should be mentioned. This method requires, as a necessary condition, knowledge of the canonical limiting values of the $^{15}N$ chemical shift of the imidazole ring of His in which each form of His is present to the extent of 100%. In this regard, there is theoretical evidence indicating that a considerable difference for the average tautomeric

equilibrium constant, $K_T$, can be obtained if DFT-computed $^{15}N$ limiting values rather than canonical limiting values are used (*Vila, 2012*), Because these results raise concerns about the magnitude of the uncertainty associated with the predictions we did not consider this method as an alternative to the above-proposed tests to validate the pKa predictions.

All in all, the estimated tautomeric forms of the imidazole ring of His are certainly not enough to accurately determine whether the coexistence of native-like and mirror-image structures occurs in solution but it could be of valuable assistance to validate the determination made by the pKa analysis.

## CONCLUSIONS

We provided a proof-of-concept of an experimental design that could be used to detect the coexistence of native and mirror-image conformations for the Q10H mutant of protein A in solution. Determination of the pKa values of the ionizable residue H10 should provide a quick answer to this problem. Additionally the NMR-determination of the one-bond vicinal coupling constant or the chemical-shifts of the imidazole ring of H10 could be used to validate this finding. There are two main advantages of the proposed methodology. Firstly, there is no need for 3D structural information and, secondly, a validation test can be carried out by standard NMR-based experiments.

Whatever the output of the proposed experiments is, we will find them interesting. Indeed, if the results do not indicate the presence of the mirror image, all the theoretical predictions about the existence of the mirror image published so far would perhaps be only of academic interest, reduced to only show a possible intermediate conformational state in the pathway of protein folding. On the other hand, if the experiments provide evidence that there is structural coexistence, then the theoretical predictions will have a sound basis and, even more important, it may spur significant progress in the conformational analysis of proteins with mirror-images.

### Funding

This research was supported by the Russian Foundation for Basic Research [15-04-00387-a], the Russian Academy of Sciences [Program 'Molecular and Cell Biology, 6.11], the MD modeling and pK calculations of a test set of proteins were supported by the Russian Science Foundation [16-14-10038] (Yury Vorobjev); the US National Institutes of Health [grant number 14312], United States National Science Foundation [grant number MCB10-19767] (Harold Abraham Scheraga); the Consejo Nacional de Investigaciones Científicas y Técnicas-Argentina (Proyectos de Investigación Plurianuales-0087], the Universidad Nacional de San Luis-Argentina (proyectos de investigación consolidados 3-2212), and by Agencia Nacional de Promoción Científica y Técnica-Argentina (PICT-0556, PICT-0767) [Jorge Alberto Vila] and (PICT-0218) [Osvaldo Antonio Martin]. The funders had no role in study design, data collection and analysis, decision to publish, or preparation of the manuscript.

## Grant Disclosures

The following grant information was disclosed by the authors:

Russian Foundation for Basic Research [15-04-00387-a].

The Russian Academy of Sciences [Program 'Molecular and Cell Biology, 6.11].

Russian Science Foundation [16-14-10038].

The US National Institutes of Health: 14312.

United States National Science Foundation: MCB10-19767.

The Consejo Nacional de Investigaciones Científicas y Técnicas-Argentina (Proyectos de Investigación Plurianuales-0087].

The Universidad Nacional de San Luis-Argentina (proyectos de investigación consolidados 3-2212).

Agencia Nacional de Promoción Científica y Técnica-Argentina (PICT-0556, PICT-0767).

## Competing Interests

The authors declare there are no competing interests.

## Author Contributions

- Osvaldo A. Martin conceived and designed the experiments, performed the experiments, analyzed the data, prepared figures and/or tables, performed the computation work, authored or reviewed drafts of the paper, approved the final draft.
- Yury Vorobjev performed the experiments, analyzed the data, authored or reviewed drafts of the paper, approved the final draft.
- Harold A. Scheraga authored or reviewed drafts of the paper, approved the final draft.
- Jorge A. Vila conceived and designed the experiments, analyzed the data, prepared figures and/or tables, authored or reviewed drafts of the paper, approved the final draft.

## Data Availability

Data is available at GitHub: https://github.com/BIOS-IMASL/protein_a_mirror.

## Supplemental Information

Supplemental information for this article can be found online at http://dx.doi.org/10.7717/peerj-pchem.2#supplemental-information.

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
