# Peer review of "Outline of an experimental design aimed to detect protein A mirror image in solution"

_PeerJ Physical Chemistry, doi:10.7717/peerj-pchem.2_

## Round 0.1 · original submission · Major Revisions

1. The main concern that has to be addressed is whether the pKa values can be predicted accurately and precisely enough to make a predicted pKa difference of 1.3 units meaningful. Evidence for and against this needs to be discussed carefully and extensively, especially in light of the short simulation time. This is the crux of the paper so a fair, balanced, and extensive discussion is needed.

2. Sufficient detail is needed regarding the MD simulations, so that the calculations can be reproduced and the reviewers questions in this regard (such a choice of protonation state for “spectator” groups) needs to be addressed.

3. There seems to be a lot of manual intervention in creating the starting structures, so the coordinates of these need to made accessible, at a minimum.

Reviewer 1 ·

Basic reporting

There are several places where a space is needed before a reference.

"MD simulationVorobjev et al."
"biologySchnell"
"task.Machuqueiro"
are three examples I found, but suggest a general check throughout.

Experimental design

According to the methods section, "As mentioned in the main text, all acid 110 (Asp, Glu) and base (Lys, Arg) residues were kept in the ionized state because their respective pKo’s 111 (3.5, 4.0, and 10.5, 12.5, respectively) are shifted by more than 2.5 pK units from the pH (7.0) at which 112 the calculations were carried out."

but pKa's can shift depending on the protein environment. Why was no check done to make sure these residues have pKas safely far away from 7? Seems like that should be straightforward given your other calculations.

Precisely how the MD simulations were set-up and processed is not detailed. Details are necessary for replication.

Validity of the findings

no comment

Additional comments

no comment

Reviewer 2 ·

Basic reporting

This paper uses computational predictions of a the structures of a histidine-containing mutant of the B-domain of staphylococcal protein A to propose an experimental test of as to whether this protein holds the generally accepted conformation or its "mirror image."

The literature review seems sufficient. I think the writing was clear, but the formatting should be improved, particularly because PeerJ uses parenthetical citations. There are several non-standard writing choices, like beginning a paragraph with "At this point is worth noting the following."

Experimental design

I believe the experimental design is too flawed for the results to be conclusive.

Also, the proposed experiment would distinguish between the conformations of a histidine-containing mutant of the protein as a function of pH. Protonation states can have a large effect on the conformation of a protein, so replacement of a glutamine residue with a histidine residue and then titrating the protonation state of the histidine.

Much of the generation of the protein conformations are done "by hand" and there is an acknowledged subjective element. e.g., "For the native conformation we used 1BDD. Gouda et al. (1992) For the mirror image we started from 2 points: a mirror-image conformation previously obtained by Vila et al, Vila et al. (2003) an a mirror image folded by hand. We discarded all the conformations obtained from this last starting point as their energies were much more higher than for the other two starting points.

The technical details of their molecular dynamics simulations are incomplete, but it seems clear that the simulations were only performed for 10 ns. Much longer simulations are necessary to calculate rigorous pKa differences. I suspect the observed differences might disappear if the simulations were performed longer. Also, pKa prediction methods of all types rarely more accurate than plus or minus one unit relative to the experimental values, so there difference of 6.1 vs 7.3 is within the margin of error of the calculations. The authors report a statistical error ~0.2-0.3 pK units, but this is likely a major underestimate due to limited sampling. At minimum, this result would have to be reproduced by other constant pH methods, such as the Brooks, Roitberg, Shen, or Roux methods. Simulation in an explicit solvent and demonstrating convergence are essential for this. Also, long-timescale simulations would be needed to show that the histidine mutant as the same conformation as the wild-type in both the A and mirror conformations as well as their relative Gibbs energies. It would also need to be shown that titration of the histidine and its fractional population in a protonated state does not induce a conformational change (which is unlikely to be true).

Validity of the findings

For the results I describe above, I do not believe it's possible to make any rigorous conclusions based on the modelling performed here.

Additional comments

My feeling is that there is not to much to be gained by further theoretical studies of this problem. Without experimental validation, it's difficult to be conclusive. The essential design of the proposed experiment introduces an amino acid that will undergo a change in charge as part of the experiment and is also fractionally populated. This could totally change the conformation of a small protein, so I don't believe the problem is well-defined. The computational pKa prediction methods would also need to be shown to be able to predict experimental values with a higher accuracy of any previous method. If this method can do that, there are many more exciting problems to study with that tool.

---

## Round 0.2 · Minor Revisions

The manuscript has improved significantly, and you need to make one small change in order for me to accept the paper. And that is a brief discussion of the following point

You write that "In addition, a large test on 297 ionized residues from 34 proteins show that a 57%, 86% and 95% of the pK prediction are with an accuracy better than 0.5, 1.0, and 1.5 pK unit respectively"

Since the error can be both positive and negative, there is a non-negligible chance that your computed values of 6.2 and 7.3 are in fact (for example) 6.7 and 6.8 respectively, i.e. there is no measurable difference in pKa. In fact there probably isn't any computational pKa prediction method that can deliver the accuracy needed to unequivocally say whether there is a measurable pKa difference. However, there is also a non-negligible chance that the pKa difference is even larger, so your work may convince some experimentalists to perform the experiment.

---

## Round 0.3 · accepted · Accept

Thank you for making the change. I am now happy to accept the manuscript for publication